# Research on Synchronous Control of Active Disturbance Rejection Position of Multiple Hydraulic Cylinders of Digging-Anchor-Support Robot

**DOI:** 10.3390/s23084092

**Published:** 2023-04-19

**Authors:** Tianbing Ma, Xiangxiang Guo, Guoyong Su, Haishun Deng, Ting Yang

**Affiliations:** 1State Key Laboratory of Mining Response and Disaster Prevention and Control in Deep Coal, Anhui University of Science and Technology, Huainan 232000, China; 2020200458@aust.edu.cn (X.G.); guoyongs005@sina.cn (G.S.); 2022201885@aust.edu.cn (T.Y.); 2College of Mechanical Engineering, Anhui University of Science and Technology, Huainan 232000, China; hsdeng@aust.edu.cn

**Keywords:** ADRC, PSO, digging-anchor-support robot, multiple hydraulic cylinders, position synchronization

## Abstract

In order to solve the problems of nonlinearity, uncertainty and coupling of multi-hydraulic cylinder group platform of a digging-anchor-support robot, as well as the lack of synchronization control accuracy of hydraulic synchronous motors, an improved Automatic Disturbance Rejection Controller-Improved Particle Swarm Optimization (ADRC-IPSO) position synchronization control method is proposed. The mathematical model of a multi-hydraulic cylinder group platform of a digging-anchor-support robot is established, the compression factor is used to replace the inertia weight, and the traditional Particle Swarm Optimization (PSO) algorithm is improved by using the genetic algorithm theory to improve the optimization range and convergence rate of the algorithm, and the parameters of the Active Disturbance Rejection Controller (ADRC) were adjusted online. The simulation results verify the effectiveness of the improved ADRC-IPSO control method. The experimental results show that, compared with the traditional ADRC, ADRC-PSO and PID controller, the improved ADRC-IPSO has better position tracking performance and shorter adjusting time, and its step signal synchronization error is controlled within 5.0 mm, and the adjusting time is less than 2.55 s, indicating that the designed controller has better synchronization control effect.

## 1. Introduction

Coal is one of the main sources of energy in China. The state attaches great importance to the development of the coal industry, and coal production is on the rise. At the same time, safety accidents have occurred frequently in our country’s coal mines, which is the issue that focuses on the safety field. The problem of coal mine safety is not being solved effectively, which is one of the key reasons limiting the rapid development of the coal industry [1,2,3]. Among the numerous coal mine safety accidents that have occurred, coal mine safety accidents caused by roof problems account for the highest proportion of total safety accidents [4]. Therefore, it is of great significance for coal mine safety to find a safe and convenient way to solve the roof problem, and roof support comes into being [5]. Digging-anchor-support robots can carry out integrated collaborative work of excavation, support and anchoring on the integrated excavation face of underground roadway to improve work efficiency [6]. The position synchronization of the multiple hydraulic cylinders of digging-anchor-support robots plays a key role in the quality of the roof support. The uncertainty and coupling effect of the multiple hydraulic cylinders of digging-anchor-support robots is more obvious than those of other multiple hydraulic cylinders. In this case, the characteristics of the hydraulic synchronous motor export to the hydraulic actuator are very different. Due to the difference in the machining and manufacturing accuracy of the hydraulic mechanism and the hydraulic motor, as well as the configuration of the hydraulic pipeline, the position synchronization error of multiple hydraulic cylinder sets will be large, which will affect the stability of the roof support [7]. Therefore, it is not only of practical guiding value but also of important safety significance to study the insufficient position synchronization accuracy of multiple hydraulic cylinders of digging-anchor-support robots under the control of a hydraulic synchronous motor.

At present, research on the position synchronization of multiple hydraulic cylinder sets of digging-anchor-support robots is still scarce. However, some achievements have been accumulated in the research on the position synchronization of multiple hydraulic cylinder groups of similar equipment and its control methods.

He [8], aiming at the problem that the stepping hydraulic support platform is difficult to balance in the support process, proposed to learn from the control strategy of the four-cylinder synchronous system by using the up-to-down bidirectional asynchronous and combining with the linear active disturbance rejection controller method to improve the position synchronization of the column cylinder when the inclination angle changes. The linear active disturbance rejection controller is convenient for parameter setting, but its control ability is weaker than that of the nonlinear active disturbance rejection controller. Lu [9] proposed a pressure-displacement closed-loop double-feedback PID control method in order to reduce the influence of stepping advanced support equipment on the stability of roof support in the process of lifting frames, which improved the position synchronization of multiple hydraulic cylinder sets. However, the PID control method has a poor control effect on nonlinear and multi-coupling systems. Wang [10] proposed an improved mean coupling control method to carry out the flow compensation strategy of electromagnetic reversing valve to eliminate position synchronization errors caused by working conditions in order to solve the problem of large synchronization errors of hydraulic synchronous motors in multiple hydraulic cylinder sets due to the differences in manufacturing accuracy and working conditions of four hydraulic cylinders. However, the control strategy and control algorithm is not combined. Wang [11] designed a support anchoring safety interlocking device combined with feedforward feedback closed-loop pump valve hydraulic system control to improve the safety and stability of anchor supporting equipment in order to prevent damage to personnel and equipment during operation and improve the stability of anchor supporting equipment. However, it increases the difficulty of equipment design, processing and control. Guo [12] proposed the use of active disturbance rejection control combined with a brainstorming optimization algorithm to adjust controller parameters online so as to ensure that the bolt rotation speed changes in real-time with different rock hardness to improve the drilling efficiency of anchor support equipment and reduce the damage rate of bolts. However, the brainstorming optimization algorithm to adjust the parameters of the active disturbance rejection controller makes it difficult to understand the principle of the algorithm and difficult to implement it. Maqsood [13] proposed a new nonlinear disturbance tracker and gained a predictive control strategy for nonlinear and uncertain disturbance problems of the system, and verified its effectiveness. However, the boundary of disturbance estimation error is difficult to determine.

Mishra [14] adopted the directional high point bidirectional asynchronous control strategy of leveling to solve the synchronization problem of multiple hydraulic cylinder sets on the weapon platform and replaced the traditional hydraulic drive system with the motor worm gear and worm gear, thus improving the position synchronization of multiple hydraulic cylinder sets on the weapon platform, but did not say that the specific control algorithm and leveling speed were slow. Xu [15] proposed an adaptive sliding mode control method based on RBF neural network in order to improve the synchronization of multiple hydraulic cylinder sets in the horizontal direction of the multi-direction die-forging hydraulic press. However, when the system reaches a steady state, due to the buffeting characteristics of the sliding mode, the stability of the platform is affected to a certain extent. Wu [16] proposed a fuzzy self-tuning integral separation PID control method, which carried out PI or PD control according to the needs of the system, and used a fuzzy algorithm to adjust variables online to reduce the tracking error and synchronization error of multi-cylinder hydraulic press. However, it was difficult to select the threshold of integral separation. Hogan [17] proposed the necessity of controlling one cylinder with one valve to improve the position synchronization performance of an electro-hydraulic servo system with multiple hydraulic cylinders, but the specific control strategies and methods are not given. Yao [18] adopted the active disturbance rejection control algorithm and extracted the Laplace matrix from the adjacent cross-coupling synchronization errors into the controller, and took the upper bandwidth limit as the standard to measure the synchronization and stability so as to realize the synchronization of the parallel electrocoating conveyor. Experiments have verified the effectiveness of the algorithm, but it is difficult to extract the Laplace matrix from the coupling synchronization errors. Xu [19] proposed a third-order active disturbance rejection controller in view of the nonlinearity and uncertainty of the six-degree-of-freedom hydraulic robot and combined the observation and compensation of the internal and external disturbance of the platform to realize the synchronous control of the position of the platform. However, the parameters of the third-order active disturbance rejection controller can hardly be optimized by human tuning. Gao [20] proposed to improve the firefly algorithm by using sine and cosine functions to accurately and quickly obtain parameters of the active disturbance rejection controller. The results show that the improved firefly algorithm improves the anti-interference ability and parameter accuracy of the active disturbance rejection controller, but the firefly algorithm has a high dependence on excellent individuals, resulting in slow convergence. In addition, it is easy to produce small amplitude oscillations near the extreme point.

Based on the analysis of the advantages and disadvantages of the above pieces of literature, an improved particle swarm active disturbance rejection control method is proposed to realize the synchronous position control of the multiple hydraulic cylinders of the anchoring robot. Firstly, the mathematical model of the multi-hydraulic cylinder platform of the anchoring robot is established; secondly, the third-order active disturbance rejection controller is designed; then, the improved particle swarm optimization algorithm is used to adjust the controller parameters. Finally, the experimental results show that the improved particle swarm active disturbance rejection controller can solve the problem of insufficient position synchronization accuracy of multiple hydraulic cylinders under the control of a hydraulic synchronous motor and has good dynamic and steady characteristics.

## 2. Modeling of the Multi Hydraulic Cylinder Group of a Digging-Anchor-Support Robot

This paper introduces the theoretical basis and process of building the simplified model and mathematical model of the multi-hydraulic cylinder system of the excavator support robot.

### 2.1. Simplified Model of Multiple Hydraulic Cylinder Groups

The multi-hydraulic cylinder system of a digging-anchor-support robot is taken as the research object, as shown in Figure 1 below. The upper part of the hydraulic legs in the multi-hydraulic cylinder set system is regarded as the load mass m and is evenly distributed on the four hydraulic legs. The eight hydraulic legs of the digging-anchor-supporting robot are simplified into four hydraulic legs, and only four hydraulic legs are stressed in actual work (four legs are stressed in rotation). Therefore, it is assumed that the four hydraulic cylinders can only move in the vertical direction along the Z axis, and the load can rotate along the X and Y axes and move vertically along the Z axis. The origin of the coordinate system is established at the center of the load, the positive direction of the X and Y-axes are parallel to both sides of the load, and the positive direction of the Z-axis is vertically upward.

Define the rolling axis Y overload center of mass, and at the same time perpendicular to the line between the piston rod of hydraulic cylinders 1, 4 and the load closing point; Define the pitch axis X overload centroid parallel to the line between the piston rod of hydraulic cylinders 1 and 4 and the load closure point, as shown in Figure 2 [21].

### 2.2. Mathematical Modeling of Multiple Hydraulic Cylinder Groups

Mechanical analysis was carried out on the simplified model of the multi-hydraulic cylinder bank system of the anchoring support robot and shown in Figure 1, and the positive direction of each variable is shown in the Figure 1. According to the force analysis, combined with Newton’s second law and the law of rigid body rotation, the dynamic equation of three degrees of freedom directions can be obtained as follows [22]:(1)$$\left\{ \begin{array}{l} \sum_{i=1}^{4}F_{i}+\sum_{i=1}^{4}f_{xi}\sin\theta_{x}+\sum_{i=1}^{4}f_{yi}\sin\theta_{y}-mg=m{\ddot{z}}_{c}\\ \sum_{i=1}^{4}\left(F_{i}+f_{xi}\sin\theta_{x}\right){\left(-1\right)}^{\sigma_{xi}}l_{xi}=J_{x}{\ddot{\theta }}_{x}\\ \sum_{i=1}^{4}\left(F_{i}+f_{yi}\sin\theta_{y}\right){\left(-1\right)}^{\sigma_{yi}}l_{yi}=J_{y}{\ddot{\theta }}_{y} \end{array}\right.$$
where:
m—load massg—acceleration due to gravity$z_{c}$—Location of the load’s center of mass$J_{x}$—moment of inertia of the load about the X-axis$J_{y}$—moment of inertia of the load about the Y-axis$\theta_{x}$ and $\theta_{y}$—the Angle of rotation about the X- and Y-axes$F_{i}$—Contact force between the I-th piston rod and the load in the Z-axis direction$f_{xi}$—Friction force between the I-th piston rod and the load in the X-axis direction$f_{yi}$—Friction force between the I-th piston rod and the load in the Y-axis direction$l_{xi}$—The length from the contact point between the ith hydraulic cylinder and the earring to the X-axis$l_{yi}$—The length from the contact point between the ith hydraulic cylinder and the earring to the Y-axis$\sigma_{xi}-f_{xi}$, $\sigma_{yi}-f_{yi}$ are the sign factors that generate the torque. When $f_{xi}$ is 0 on the left side of the X-axis and 1 on the right side, it is also true that $f_{yi}$ is 0 on the left side of the Y-axis and 1 on the right side.

### 2.3. Mathematical Model of Valve-Controlled Cylinders

#### 2.3.1. Mathematical Model for Displacement of Valve Core of Electro-Hydraulic Proportional Valve

When the electro-hydraulic proportional valve is working, the electromagnet in the valve produces the corresponding action according to the input voltage signal, which causes the valve core to shift. In this paper, the mathematical model of the electro-hydraulic proportional valve is regarded as the first-order proportional link by default, and its transfer function is as follows:(2)$$K_{v}=\frac{X_{v}}{u}$$
where *X_v_* is the displacement of the valve core of the electro-hydraulic proportional valve, *K_v_* is the displacement gain of the electro-hydraulic proportional valve, and *u* is the input voltage of the electro-hydraulic proportional valve.

#### 2.3.2. Flow Equation of Electro-Hydraulic Proportional Valve

To simplify the analysis, we make the following assumptions:(1)Each orifice of the slide valve is symmetrically distributed;(2)Hydraulic fluid is in the ideal condition;(3)The electrohydraulic proportional valve has good response-ability.

According to this, the flow equation of the electro-hydraulic proportional valve can be written as:(3)$$Q_{L}=K_{q}X_{v}-K_{c}p_{L}$$
where *K_q_* is the flow gain of the electro-hydraulic proportional valve, *K_c_* is the flow rate of the electro-hydraulic proportional valve-is the pressure coefficient, and *p_L_* is the hydraulic cylinder control chamber pressure.

#### 2.3.3. Hydraulic Cylinder Flow Continuous Equation

To simplify the analysis, we make the following assumptions:(1)The pipes connecting components have a large cross-sectional area and ignore the pressure drop caused by the flow of hydraulic fluid in the pipes;(2)The pressure applied to the default working chamber is average, and the viscosity difference of the hydraulic fluid due to temperature changes is ignored.

From the above, the flow equation of the hydraulic cylinder control chamber is as follows:(4)$$q_{L}=A_{1}\frac{dx_{p}}{dt}+\frac{V_{1}}{\beta_{e}}\frac{dp_{1}}{dt}-C_{ip}\left(p_{2}-p_{1}\right)$$
where *C_ip_* is the leakage coefficient in the valve, *V*_1_ is the volume of the control chamber in the cylinder, *A*_1_ is the area of the rodless chamber, *p*_1_ is the pressure of the rodless chamber, *p*_2_ is the pressure of the rodless chamber, *V*_1_ = 2*LA*_1_, and *L* is the maximum stroke of the hydraulic cylinder.

In addition, there are:(5)$$V_{1}=V_{0}+A_{1}x_{p}$$
where *V*_0_ is the initial volume of the hydraulic cylinder control chamber.

Assuming that the piston displacement is small enough, that is |*A*_1_*x_p_*| < <*A*_0_, *V*_1_ is approximately equal to *V*_0_, according to the comprehensive Formulas (4) and (5):
(6)$$q_{L}=A_{1}\frac{dx_{p}}{dt}+\frac{V_{0}}{\beta_{e}}\frac{dp_{1}}{dt}-C_{ip}\left(p_{2}-p_{1}\right)$$

Its incremental Laplace transform is:(7)$$Q_{L}=A_{1}sX_{p}+C_{ip}p_{1}+\frac{V_{0}}{\beta_{e}}sP_{1}$$

#### 2.3.4. Dynamic Balance Equation of Hydraulic Cylinder Load

The load dynamic balance equation of the supporting hydraulic cylinder can be obtained as follows:(8)$$p_{1}A_{1}-p_{2}A_{2}=m_{t}\frac{d^{2}x_{p}}{dt^{2}}+B_{p}\frac{dx_{p}}{dt}+Kx_{p}+F_{L}$$
where *m_t_* is the load mass, *B_p_* is the viscous damping coefficient of the piston and load, *K* is the load spring stiffness and *F_L_* is the static load force on the piston.

The Laplace transform equation for its increment is:(9)$$p_{1}A_{1}=m_{t}s^{2}X_{p}+B_{p}sX_{p}+Kx_{p}+F_{L}$$

#### 2.3.5. Transfer Function of Valve-Controlled Hydraulic Cylinder

Simultaneous Formulas (10)–(12):
(10)$$Q_{L}=K_{q}X_{v}-K_{c}p_{L}$$
(11)$$Q_{L}=A_{1}sX_{p}+C_{ip}p_{1}+\frac{V_{0}}{\beta_{e}}sP_{1}$$
(12)$$p_{1}A_{1}=m_{t}s^{2}X_{p}+B_{p}sX_{p}+Kx_{p}+F_{L}$$

The output displacement function of the piston can be obtained by eliminating the variable *Q_L_*, *p*_1_, *p*_2_ as follows:(13)$$X_{p}=\frac{\frac{K_{q}}{A_{1}}X_{v}-\frac{K_{ce}}{A_{1}^{2}}\left(1+\frac{V_{0}}{\beta_{e}K_{ce}}s\right)F_{L}}{\frac{V_{0}m_{t}}{\beta_{e}A_{1}^{2}}s^{3}+\left(\frac{m_{t}K_{ce}}{A_{1}^{2}}+\frac{B_{p}V_{0}}{\beta_{e}A_{1}^{2}}\right)s^{2}+\left(1+\frac{B_{p}K_{ce}}{A_{1}^{2}}+\frac{KV_{O}}{\beta_{e}A_{1}^{2}}\right)s+\frac{K_{ce}K}{A_{1}^{2}}}$$
where *K_ce_* is the total pressure flow coefficient and *K_ce_ = K_c_ + C_tp_*

According to the above formula, the transfer function of the displacement *X_p_* of the piston rod relative to the displacement *X_v_* of the valve core is:(14)$$\frac{X_{p}(s)}{X_{v}(s)}=\frac{\frac{K_{q}}{A_{1}}}{\frac{V_{0}m_{t}}{\beta_{e}A_{1}^{2}}s^{3}+\left(\frac{m_{t}K_{ce}}{A_{1}^{2}}+\frac{B_{p}V_{0}}{\beta_{e}A_{1}^{2}}\right)s^{2}+\left(1+\frac{B_{p}K_{ce}}{A_{1}^{2}}+\frac{KV_{o}}{\beta_{e}A_{1}^{2}}\right)s+\frac{K_{ce}K}{A_{1}^{2}}}$$

In the process of the above analysis, we establish the mathematical model of the core displacement of the proportional valve, the flow equation of the proportional valve, the flow continuity equation of the hydraulic cylinder and the load dynamic balance equation, and derive the transfer function of the valve-controlled hydraulic cylinder. The relationship between the command u and the displacement of the hydraulic cylinder is proved, that is, the command u is inputted into the electro-hydraulic proportional valve, and the valve core is displaced by the electromagnet in the valve according to the input signal. The displacement of the piston rod of the hydraulic cylinder is controlled by controlling the opening size of the valve to control the flow rate. The mathematical model of the controlled object is improved, which provides theoretical support for the later experimental design of multiple hydraulic cylinder group controllers for digging-anchor-support robots.

## 3. Design of Multiple Hydraulic Cylinder Group Controller for Digging-Anchor-Support Robot

This paper introduces three important components of active disturbance rejection controller structure for multiple hydraulic cylinders of digging anchor support robots and how to improve particle swarm optimization algorithm to adjust and optimize the parameters of active disturbance rejection controller.

### 3.1. Structure Design of Active Disturbance Rejection Controller

Active disturbance rejection control technology is a new modern control theory built by Professor Han Jingqing on the basis of a PID controller. It is characterized by the design of the control structure of uncertain, disturbed and nonlinear systems from tedious to simple, from abstract to easy to understand [23,24].

The basic control structure of ADRC is mainly composed of three parts: tracking differentiator, expanded state observer and nonlinear state error feedback control law [25], as shown in Figure 3:

### 3.2. Design of Active Disturbance Rejection Controller Algorithm

#### 3.2.1. Tracking Differentiator

The TD module in Figure 3 is an active disturbance rejection tracking differential controller whose function is to make the output signal change in proportion to the input signal as quickly as possible to avoid the impact of signal fluctuations on the system. The active disturbance rejection differential tracker is adopted, which has a low delay and good tracking performance. The specific expression is as follows [26]:(15)$$\left\{ \begin{array}{l} fh=fhan\left(x_{1}\left(k\right)-v\left(k\right),x_{2}\left(k\right),r\left(h\right)\right)\\ x_{1}\left(k+1\right)=x_{1}\left(k\right)+Tx_{2}\left(k\right)\\ x_{2}\left(k+1\right)=x_{2}\left(k\right)+Tf\left(h\right) \end{array}\right.$$
(16)$$u=fhan\left(x_{1},x_{2},r,h_{0}\right):$$
(17)$$\left\{ \begin{array}{l} d=rh\\ d_{0}=hd\\ y=x_{1}+hx_{2}\\ a_{0}=\sqrt{d^{2}+8r\left|y\right|}\\ a=\left\{ \begin{array}{cc} x_{2}+\frac{\left(a_{0}-d\right)}{2}sign\left(y\right) & ,\left|y\right|>d_{0}\\ x_{2}+\frac{y}{h} & ,\left|y\right|\leq d_{0} \end{array}\right.\\ fhan=-\left\{ \begin{array}{cc} rsign\left(a\right) & ,\left|a\right|>d\\ r\frac{a}{d} & ,\left|a\right|\leq d \end{array}\right. \end{array}\right.$$
where *h* is the sampling period, *r* is the velocity factor and *h*_0_ is the filter factor function. The expression is shown in Formula (17).

The larger the *r* value is, the shorter the transition time will be. However, too much makes the transition process meaningless; the larger the *h*_0_ value is, the better the noise removal effect will be. However, the larger the *h*_0_ value is, the greater the output signal will be distorted. According to the experience of our predecessors, set *h*_0_ 3–10 h.

#### 3.2.2. Extended State Observer

The basic idea of the extended observer is to observe and estimate the displacement, velocity, acceleration and disturbance of the input signal and feedback to the controller in real time. The discrete algorithm of the third-order nonlinear extended state observer is as follows [27]:(18)$$\{\begin{array}{l} e=z_{1}(k)-y\\ z_{1}(k+1)=z_{1}(k)+h\left(z_{2}(k)-\beta_{01}e\right)\\ z_{2}(k+1)=z_{2}(k)+h\left(z_{3}(k)-\beta_{02}fal\left(e,a_{01},\sigma_{}\right)\right)\\ z_{3}(k+1)=z_{3}(k)+h\left(z_{4}(k)-\beta_{03}fal\left(e,a_{02},\sigma_{}\right)+b_{0}u(k)\right)\\ z_{4}(k+1)=z_{4}(k)-h\beta_{04}fal\left(e,a_{03},\sigma_{}\right) \end{array}$$
(19)$$\mathrm{fal}\left(e,a,\sigma_{}\right)=\{\begin{array}{ll} \frac{e}{\sigma^{1-\alpha }}, & |e|⩽\sigma \\ |e{|}^{\alpha }\mathrm{sign}(e), & |e|>\sigma \end{array}$$
where *y* and *u* are inputs, $z_{1},z_{2},z_{3},z_{4}$ are observation and estimation signals, $\beta_{01},\beta_{02},\beta_{03},\beta_{04}$ are observer gain factors, $a_{01},a_{02},a_{03}$ are observer nonlinear coefficients and $b_{0}$ are compensation coefficients. Generally, according to experience, $a_{01},a_{02},a_{03}$ are taken as 0.5, 0.25, 0.125.

#### 3.2.3. Nonlinear State Error Feedback Control Law

The nonlinear state error feedback control law combines the difference between the output signals of the tracking differentiator and the extended observer with the nonlinear function and calculates the output control quantity u. Its discrete algorithm is as follows [28]:(20)$$\{\begin{array}{l} e_{1}=v_{1}(k)-z_{1}(k)\\ e_{2}=v_{2}(k)-z_{2}(k)\\ e_{3}=v_{3}(k)-z_{3}(k)\\ u_{0}=\beta_{11}fal\left(e_{1},a_{11},\sigma_{}\right)+\beta_{12}fal\left(e_{2},a_{12},\sigma_{}\right)+\\ \;\beta_{13}fal\left(e_{3},a_{13},\sigma_{}\right)\\ u(k)=u_{0}-z_{4}(k)/b_{0} \end{array}$$

Generally, $0<a_{11}<1<a_{12}<a_{13}$, 0.75, 1.25, 1.50 or −0.6, 0.6, 1.2, depending on experience; larger is more likely to lose the advantage of nonlinear gain, and smaller systems are more unstable; generally, take 0.01<σ<0.1.

### 3.3. Parameter Optimization of Active Disturbance Rejection Controller Based on Improved Particle Swarm Optimization

#### 3.3.1. Standard Particle Swarm Optimization

Standard particle swarm optimization has the characteristics of a simple control principle, strong practicability and a wide application range. The evolution updating formula of the standard particle swarm optimization algorithm is as follows [29,30]:(21)$$\{\begin{array}{l} v_{id}(k+1)=\omega v_{id}(k)+c_{1}r_{1}\left(p_{id}-z_{id}(k)\right)+c_{2}r_{2}\left(p_{gd}-z_{id}(k)\right)\\ z_{id}(k+1)=z_{id}(k)+v_{id}(k+1) \end{array}$$
where *k* is the number of current iterations; *z_i_* is the i particle; *d* is the dimension of the particle; *p_i_* is the optimal historical value; *p_g_* is the optimal global value; *c*_1_ and *c*_2_ are acceleration constants; *w* is particle inertia weight; *r*_1_ and *r*_2_ are mutually independent random numbers in [0, 1].

#### 3.3.2. Improvement of Algorithm

The convergence speed of standard particle swarm optimization is slow, and it is easy to fall into local optimal. The compression factor and the theoretical results of the genetic algorithm are used to improve the traditional particle swarm optimization algorithm to improve the optimization range and convergence speed [31]. The velocity updating formula of the compression factor method is:(22)$$v_{id}(k)=\lambda v_{id}(k)+c_{1}r_{1}\left(p_{id}-z_{id}(k)\right)+c_{2}r_{2}\left(p_{gd}-z_{id}(k)\right)$$
where λ is the compression factor, and its expression is as follows:(23)$$\lambda =\frac{2}{\left|2-\varphi -\sqrt{\left(\varphi^{2}-4\varphi \right)}\right|}$$
(24)$$\varphi =c_{1}+c_{2}$$

After the introduction of the compression factor, the standard particle swarm optimization algorithm converges faster and has the characteristics of simple structure and strong applicability. However, due to the increasing number of iterations, the optimal solution may be formed locally and unable to jump out, and the global search ability is weak. In order to solve this problem, the essence of the genetic optimization algorithm is referred to, and the crossover and selection process of the genetic algorithm is added into the process of iterative optimization to improve and optimize the particle swarm optimization algorithm so as to obtain a larger scope of optimization.

The details are as follows: calculate the arithmetic mean value of the fitness function of each generation of particles, take an even number of particles whose fitness function value is greater than the arithmetic mean value and send them into the next iteration optimization, and use the process of selection and crossover in genetic algorithm to randomly select the number of particles that need to be crossed. These particles are cross-matched to form offspring, and their fitness function values are calculated and compared with those of the previous parent generation. The better fitness function values are selected and sent to the next iteration of particle optimization to generate a new set of particles.

ITAE error criterion is selected as the fitness function of the improved optimization algorithm [32], and the standard form is as follows:(25)$$J=\int_{0}^{\infty }t|e(t)|dt$$

There are 18 unknown parameters in the designed third-order nonlinear active disturbance rejection controller for the multi-hydraulic cylinder system of the digging-anchor-support robot, in which the observer gain factors $\beta_{01},\beta_{02},\beta_{03},\beta_{04}$ in the extended observer and the nonlinear function gain $\beta_{11},\beta_{12},\beta_{13},b_{0}$ in the nonlinear state error feedback control law are eight important parameters for the satisfactory performance of the designed controller. The improved particle swarm optimization algorithm takes the above eight important parameters as particles to optimize the third-order nonlinear active disturbance rejection controller of the multi-hydraulic cylinder system of the excavator support robot.

The specific process of the improved algorithm is shown in Figure 4.

## 4. Simulation Analysis

In order to verify the reliability of the ADRC-IPSO control algorithm, compared with the traditional auto-disturbance rejection control algorithm and the classical PID control algorithm, the tracking speed is faster and the tracking performance is better. The joint simulation is carried out by using MATLAB and AMESim software. In the simulation experiment, the step signal and sine signal are used as the desired signal, as shown in Table 1.

### 4.1. System Physical Simulation Model

When the simulation is running, the displacement sensor built in the AMESim software sends the collected displacement signal to the common simulation interface, and the multi-hydraulic cylinder group system ADRC reads the S function generated by the internal operation of the common simulation interface, and then the valve core displacement The closed-loop control signal is fed back to the AMESim software to adjust the displacement of the supporting hydraulic cylinder in real-time so as to ensure the synchronous movement of the four supporting hydraulic cylinders equipped with the multi-hydraulic cylinder group of the digging-anchor-support robot. The AMESim simulation model of the multi-hydraulic cylinder group equipment position synchronization control system of the digging-anchor-support robot is shown in Figure 5.

### 4.2. Simulation Analysis When the Expected Signal Is a Step Signal

#### 4.2.1. Displacement Tracking Performance Simulation Experiment

When the simulation time is 10 s, and the signal amplitude is 150 mm, the displacement tracking performance of the multi-hydraulic cylinder group of the digging-anchor-support robot is compared with traditional ADRC, ADRC-PSO and classical PID control algorithm under the ADRC-IPSO algorithm. The results are shown in Figure 6, Figure 7, Figure 8 and Figure 9. 

It can be seen from Figure 6, Figure 7, Figure 8 and Figure 9 and Table 2: in the simulation, the four-cylinder displacement does not overshoot under the four control algorithms of PID, ADRC, ADRC-PSO and ADRC-IPSO for the step signal with amplitude 150 mm. The steady-state points of the displacement curves of the four hydraulic cylinders are shown as stars in the figure, and their adjustment time and steady-state values have been marked in the figure. Compared with the PID control, the minimum adjustment time of the step signal under the control of ADRC-IPSO is reduced by 63.6%, the maximum adjustment time is reduced by 62.3%, and the tracking process is faster. This is because the tracking differentiator and extended observer in the auto disturbance rejection controller improve the response speed of the system and the observation and compensation of the system state variables. Compared with ADRC, the minimum adjustment time of step signal under the control of ADRC-IPSO is reduced by 51.1%, and the maximum adjustment time is reduced by 56.2%. This is due to the introduction of an improved particle swarm optimization algorithm in ADRC-IPSO, which optimizes many parameters of the controller in real-time, avoids manual tuning of parameters, and improves the reliability of the controller. Compared to ADRC-PSO control, under the control of ADRC-IPSO, the minimum adjustment time of the displacement curves of the four support hydraulic cylinders when reaching a stable state is reduced by 39.7%, and the maximum adjustment time is reduced by 41.8%. This is because the improved particle swarm auto-disturbance rejection controller uses compression factors instead of inertia weights and draws on the crossover and selection processes in genetic algorithms to improve the optimization range of traditional particle swarm algorithms. To avoid falling into local and global optimization, it is possible to find suitable parameters for the multi-hydraulic cylinder group equipment of the digging-anchor-support robot to adjust the movement of the four supporting hydraulic cylinders, ensuring the position synchronization of the four hydraulic cylinders.

Finally, given the step signal, the minimum adjustment time for the displacement curves of the four support hydraulic cylinders under the control of ADRC-IPSO when reaching a stable state is 0.44 s, and the maximum adjustment time is 0.46 s.

#### 4.2.2. Synchronization Error Simulation Experiment

The simulation experiment of synchronization error of the multi-hydraulic cylinder group of the digging-anchor-support robot under the ADRC-IPSO algorithm, traditional ADRC algorithm, ADRC-PSO algorithm and classical PID algorithm is carried out. The results are shown in Figure 10, Figure 11, Figure 12 and Figure 13.

It can be seen from Figure 10, Figure 11, Figure 12 and Figure 13 and Table 3 that under the expected signal of amplitude 150 mm, the synchronization errors of four cylinders fluctuate in different ranges under the control of PID, ADRC, ADRC-PSO and ADRC-IPSO respectively, but all tend to zero. The subfigures in the figure extracts the synchronization error curve of the hydraulic cylinder within 0–1.5 s and amplifies it. The maximum synchronization error points of the four hydraulic cylinders are shown as stars in the figure, and their values have been marked in the figure. Compared with PID control, the minimum synchronization error of the platform under ADRC-IPSO control is reduced by 64.9%, and the maximum synchronization error is reduced by 57.5%. Compared with ADRC control, the minimum synchronization error of the platform under ADRC-IPSO control is reduced by 50.4%, and the maximum synchronization error is reduced by 44.9%. Compared with the ADRC-PSO control, the maximum synchronization error of the multi-hydraulic cylinder group position synchronization control system under ADRC-IPSO control is reduced by 37.8%, and the minimum synchronization error is increased by 34.3%. The absolute value of the maximum synchronization error of the multi-hydraulic cylinder group position synchronization control system is 2.93 mm, which satisfies the condition that the maximum synchronization error of the system supports 4% of the stroke of the hydraulic cylinder. This is because the particle swarm optimization algorithm in the ADRC-IPSO algorithm uses a selection and crossover process of genetic algorithms, allowing it to find more suitable parameters to adjust the movement position of the piston rod of the supporting hydraulic cylinder in real-time compared to other controllers, Adapting to the nonlinearity, coupling and uncertainty in the operation of the multi-hydraulic cylinder group equipment of the digging-anchor-support robot.

Finally, given the step signal, the maximum position synchronization error of the four support hydraulic cylinders during movement under the control of ADRC-IPSO is 2.93 mm, and the minimum position synchronization error is −1.88 mm.

### 4.3. Simulation Analysis When the Expected Signal Is Sinusoidal

#### 4.3.1. Displacement Tracking Performance Simulation Experiment

Setting the time of each experiment as 10 s, the signal amplitude as 150 mm and the frequency as 1 rad/s, the displacement tracking performance of the multi-hydraulic cylinder group of digging-anchor-support robots under the ADRC-IPSO algorithm, traditional ADRC algorithm, ADRC-PSO algorithm and classical PID control algorithm are compared. The results are shown in Figure 14, Figure 15, Figure 16 and Figure 17.

It can be seen from Figure 14, Figure 15, Figure 16 and Figure 17 and Table 4, under the four control algorithms of PID, ADRC, ADRC-PSO and ADRC-IPSO, although the displacement of the four cylinders fluctuates with different amplitudes in the four control algorithms of 150 mm and 1 rad/sec, the displacement of the four cylinders finally tends to be stable, and when the tracking displacement curve reaches the amplitude position, there is a certain amplitude error between the tracking displacement curve and its expected value due to the shock and vibration of the hydraulic oil. In the PID control algorithm, the three parameters are linearly combined, and the tracking effect of the input signal with the expected value varying with time is not ideal, so the tracking displacement curve controlled by the PID algorithm fluctuates greatly when the sine signal is used as the expected signal, and the tracking curve under the ADRC control fluctuates greatly within 0.75 s, but the tracking effect is good, which is caused by the improper manual setting parameters of the auto-disturbance rejection controller. The tracking displacement curve of the four cylinders under the control of the ADRC-PSO algorithm fluctuates slightly up and down, but due to the particle swarm optimization algorithm easily falling into local optimization, no suitable parameters are found, and the maximum fluctuation error is large. The subfigures in the figure extracts the tracking error curve of the hydraulic cylinder within 0–0.5 s and amplifies it. The maximum tracking error point of the hydraulic cylinder is shown as a star in the figure, and its value has been marked in the figure. Compared with PID control, the maximum fluctuation error of the platform under ADRC-IPSO control is reduced by 43.4%, the maximum amplitude error is reduced by 80.6%, the maximum adjustment time is reduced by 80.3%, and the minimum adjustment time is reduced by 78.2%. Compared with ADRC control, the maximum fluctuation error of the platform under ADRC-IPSO control is reduced by 33.7%, the maximum amplitude error is reduced by 34.5%, the maximum adjustment time is reduced by 50.7%, and the minimum adjustment time is reduced by 21.6%. Compared to ADRC-PSO control, the maximum fluctuation error of the multi-hydraulic cylinder group equipment of the digging-anchor-support robot under ADRC-IPSO control has been reduced by 26.4%, the maximum amplitude error has been reduced by 13.6%, the maximum adjustment time has been reduced by 54.4%, and the minimum adjustment time has been reduced by 40.8%. Finally, the maximum fluctuation error, maximum amplitude error, maximum adjustment time, and minimum adjustment time of the multi-hydraulic cylinder group equipment of the digging-anchor-support robot under the ADRC-IPSO control algorithm under a given sinusoidal signal are −6.9 mm, −1.9 mm, 0.37 s, and 0.29 s, respectively.

#### 4.3.2. Synchronization Error Simulation Experiment

The simulation experiment of synchronization error comparison between the ADRC-IPSO control algorithm, the ADRC control algorithm, the ADRC-PSO control algorithm and the classical PID control algorithm is carried out for a multi-hydraulic cylinder group of digging-anchor-support robot. The results are shown in Figure 18, Figure 19, Figure 20 and Figure 21:

It can be seen from Figure 18, Figure 19, Figure 20 and Figure 21 and Table 5 that under the expected signal of amplitude 150 mm and frequency 1 rad/sec, and under the control of PID, ADRC, ADRC-PSO and ADRC-IPSO algorithms, the four-cylinder synchronization error curve of the four-cylinder hydraulic synchronization platform has different amplitude and frequency fluctuations. The maximum synchronization error points of the four hydraulic cylinders are shown as stars in the figure, and their values have been marked in the figure Compared with the PID control, the minimum synchronization error under the ADRC-IPSO control is reduced by 67.7%, the maximum synchronization error is reduced by 51.9%, and the stability error is reduced by 93.1%. Compared with the ADRC control, the minimum synchronization error under the ADRC-IPSO control is reduced by 55.3%, the maximum synchronization error is reduced by 33.9%, and the stability error is reduced by 83.3%. Compared to the ADRC-PSO control algorithm, the minimum synchronization error of the multi-hydraulic cylinder group system under ADRC-IPSO control is reduced by 32.7%, the maximum synchronization error is reduced by 16.3% and the stability error is reduced by 75.3%. The maximum synchronization error of the multi-hydraulic cylinder group system is 3.9 mm, which meets the experimental purpose of supporting 4% of the hydraulic cylinder stroke with the synchronization error. During the entire experimental process, compared to ADRC-IPSO, the tracking effects of PID, ADRC and ADRC-PSO are not ideal, with large fluctuations and limited control capabilities for a large range of tracking signal changes. The ADRC-IPSO controller uses a particle swarm optimization algorithm to find more suitable parameters for the system to adjust the fluctuations generated during the tracking process and has achieved good results.

Finally, when the multi-hydraulic cylinder group of the digging-anchor-support robot is equipped with a given sinusoidal signal, the maximum synchronization error of the hydraulic cylinder under ADRC-IPSO control is −3.9 mm, the minimum synchronization error is 2.1 mm, and the stability error is 0.2 mm.

## 5. Experimental Analysis

From the above theoretical simulation analysis, we can see that in the position synchronization control of the multi-hydraulic cylinder group system of the digging-anchor-support robot, the ADRC-IPSO algorithm is more effective than the traditional ADRC algorithm and the classical PID control algorithm, the tracking speed is faster and the tracking performance is better, which verifies the reliability of the controller and provides a basis for the following experiments.

### 5.1. Experimental Platform Parameters

Taking the multi-hydraulic cylinder group system of the digging-anchor-support robot as the research object, the inner diameter of the hydraulic cylinder, piston rod diameter and stroke are 80 mm, 50 mm and 150 mm, respectively. The relevant experimental parameters are shown in Table 6.

### 5.2. ADRC Controller Parameter Settings

From a large number of experiments and engineering application experience, it is known that the parameter setting of the ADRC of the system plays a key role in the effect of the experiment and the result of the engineering application. Read the relevant literature and learn from the methods, set the parameters of the auto disturbance rejection controller, and use the improved particle swarm optimization algorithm to optimize it. Through a large number of simulation analyses of the multi-hydraulic cylinder group system of the digging-anchor-support robot, set the optimization parameters of ADRC-IPSO, as shown in Table 7.

### 5.3. Introduction to Experimental Hardware

#### 5.3.1. Displacement Sensor

In this paper, the Witte Smart WT53R-485 laser displacement sensor is selected, the power supply voltage is 5–6 V, the baud rate is optional in the range of 2,400,921,600, the default is 115,200 and the measurement range is 4–400 cm. In order to ensure the reliability of the experimental data, during the experiment, try to keep the surface of the measured object parallel to the laser emitting surface of the laser ranging sensor. Since the laser light source is an invisible beam-type laser, it is necessary to ensure that on the surface of the measured object, there are no obstacles and fine particles within the angle.

#### 5.3.2. Electric Control System and Operation Platform

For the complex multi-hydraulic cylinder equipment of the digging-anchor-support robot, the CSPACE semi-physical simulation control system is used to build an accurate system control model, and the joint debugging test is repeated and improved until each hydraulic unit can work normally. At the same time, based on the MBD method, the ADRC algorithm is optimized and improved, and the core problems, such as the parameter identification of the ADRC controller and the position synchronization of multiple hydraulic cylinder groups and multiple cylinders, are efficiently solved. The electrical control system and operating platform of the multi hydraulic cylinder group equipment of the excavation anchor support robot are shown in Figure 22.

Before the experiment starts, it is necessary to configure certain parameters of the experimental hardware, debug the equipment, set the CAN_A baud rate to 1 M, the permanent magnet synchronous motor driver RS485 communication baud rate to 28,800 and the upper computer RS232 communication baud rate to 115,200.

### 5.4. ADRC Process of Hydraulic Cylinder Groups for the Digging-Anchor-Support Robot 

The multi-hydraulic cylinder group experimental platform of the digging-anchor-support robot is composed of a permanent magnet synchronous motor, small hydraulic fuel tank, solenoid valve, hydraulic synchronous motor, asymmetric hydraulic cylinder, controller, laser ranging sensor and electric control cabinet. In the experiment, the small hydraulic oil tank is driven by the permanent magnet synchronous motor to transport the hydraulic oil to the hydraulic synchronous motor, the hydraulic oil is diverted, and the controller controls the hydraulic oil flow into the hydraulic cylinder by controlling the opening size of the electromagnetic directional valve.

In the experiment, the small hydraulic oil tank is driven by the permanent magnet synchronous motor to transport the hydraulic oil to the hydraulic synchronous motor, the hydraulic oil is diverted, and the controller controls the hydraulic oil flow into the hydraulic cylinder by controlling the opening size of the electromagnetic directional valve. The laser range sensor above the hydraulic cylinder side transmits the collected displacement signal to the controller, and the controller uploads the collected data to the upper computer of the electric control cabinet for display and adjusts the opening size of the solenoid valve according to the displacement data, so as to ensure the synchronization of the multi-hydraulic cylinder group and ensure that the ADRC experiment of the digging-anchor-support robot can be carried out smoothly, as shown in Figure 23 and Figure 24.

### 5.5. Analysis of Experimental Results

#### 5.5.1. Experimental Analysis of Displacement Tracking Performance of Each Cylinder

Setting the time of each experiment as 10 s and the expected value as 150 mm, the displacement tracking performance of the multi-hydraulic cylinder group of the digging-anchor-support robot is compared with that of improved particle swarm auto-disturbance rejection control algorithm, traditional auto-disturbance rejection control algorithm and classical PID control algorithm. The results are shown in Figure 25, Figure 26 and Figure 27: 

It can be seen from Figure 25, Figure 26 and Figure 27 and Table 8 that although the step signal with 150 mm amplitude is controlled by PID, ADRC and ADRC-IPSO control algorithms, the four cylinders do not overshoot in the process of displacement tracking, but they all fluctuate in a small range up and down the steady state value after reaching the expected value. This is because the impact and vibration of the hydraulic cylinder cause interference to the system after the hydraulic cylinder moves to the predetermined position. The subfigures in the figure extracts the displacement curve of the hydraulic cylinder within 2–5 s and amplifies it. The steady-state points of the hydraulic cylinder displacement curve are shown as stars in the figure, and their adjustment time and steady-state values have been marked in the figure. Under the control of the ADRC-IPSO algorithm, the minimum adjustment time of the step signal is reduced by 47.0%, and the maximum adjustment time is reduced by 44.7% compared with that under the PID control, and the tracking process is smoother. This is due to the fact that in the auto disturbance rejection control part of ADRC-IPSO, the tracking differentiator and extended observer improve the system response speed and the observation and compensation of system state variables. The minimum adjustment time of the step signal under the control of the ADRC-IPSO algorithm is 36.4% less than that under the ADRC control, and the maximum adjustment time is reduced by 37.2%. This is due to the introduction of the improved particle swarm optimization algorithm into the traditional auto disturbance rejection controller, which optimizes many parameters processed by the controller in real-time, avoids manual tuning of parameters and improves the reliability of the controller.

#### 5.5.2. Experimental Analysis of Displacement Synchronization Error of Each Cylinder

Experiments were carried out to compare the synchronization errors of the multi-hydraulic cylinder group of the digging-anchor-support robot under the improved particle swarm auto-disturbance rejection control algorithm, the traditional auto-disturbance rejection control algorithm and the classical PID control algorithm. The results are shown in Figure 28, Figure 29 and Figure 30:

It can be seen from Figure 28, Figure 29 and Figure 30 and Table 9 that the synchronization errors of the four cylinders of the multi-hydraulic cylinder group platform of the digging-anchor-support robot fluctuate in different ranges under the control of PID, ADRC and ADRC-IPSO, respectively. The maximum synchronization error points of the four hydraulic cylinders are shown as stars in the figure, and their values have been marked in the figure Among them, under the control of the ADRC-IPSO algorithm, the minimum synchronization error is reduced by 55.1%, the maximum synchronization error is reduced by 43.2%, and the steady-state error is reduced by 0.2 mm. The minimum synchronization error under ADRC-IPSO control is 38.0% lower than that under ADRC control, and the maximum synchronization error is reduced by 38.3%. The maximum synchronization error of 5 mm of the platform satisfies the experimental condition that the synchronization error is less than or equal to 4% of the hydraulic cylinder stroke. Because ADRC-IPSO adopts the selection and crossover process of the genetic algorithm, it can find more suitable parameters for the auto disturbance rejection controller in real-time to adapt to the nonlinearity, coupling and uncertainty in the operation of the multi-hydraulic cylinder group platform of the digging-anchor-support robot.

## 6. Conclusions

(1)Taking the multi-hydraulic cylinder group system of the digging-anchor-support robot as the research object, due to the lack of four-cylinder position synchronization accuracy of the system under the control of a hydraulic synchronous motor, an improved particle swarm auto-disturbance rejection control algorithm is proposed on the basis of building the mathematical model of the multi-hydraulic cylinder group system of the digging-anchor-support robot. Its step signal synchronization error and adjusting time are controlled within 5.0 mm and 2.55 s, respectively. Compared with the traditional ADRC algorithm, it is reduced by 38.3% and 37.2%, respectively;(2)The simulation results show that after introducing the crossover and selection operation of the genetic algorithm, compared with the traditional ADRC algorithm, the adjustment time under the action of a step signal and a sinusoidal signal is reduced by 56.2% and 80.3% respectively, and the synchronization error is reduced by 44.9% and 30.9%. Compared to traditional particle swarm optimization active disturbance rejection control algorithms (ADR-PSO), the adjustment time is reduced by 41.8% and 54.4%, respectively, under the action of step signals and sinusoidal signals, and the position synchronization error is reduced by 37.8% and 16.3% respectively, which verifies the feasibility of the improved PSO algorithm;(3)The experimental results show that compared with ADRC and PID controllers, ADRC-IPSO has better displacement tracking performance, smaller synchronization error and faster reaching steady state without overshoot, and the improved particle swarm optimization algorithm better optimizes the parameters of ADRC controller and makes it easier to implement and has better dynamic and steady-state characteristics. It also solves the problem of insufficient position synchronization accuracy of the multi-hydraulic cylinder group system of the digging-anchor-support robot under the control of a hydraulic synchronous motor.

## Figures and Tables

**Figure 1 sensors-23-04092-f001:**
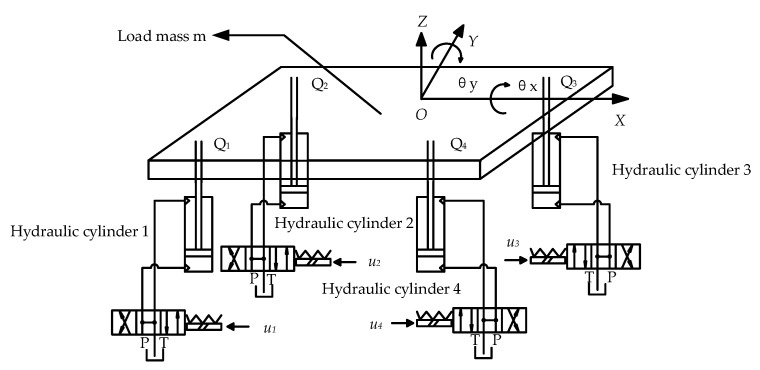
Simplified model of multiple hydraulic cylinder groups.

**Figure 2 sensors-23-04092-f002:**
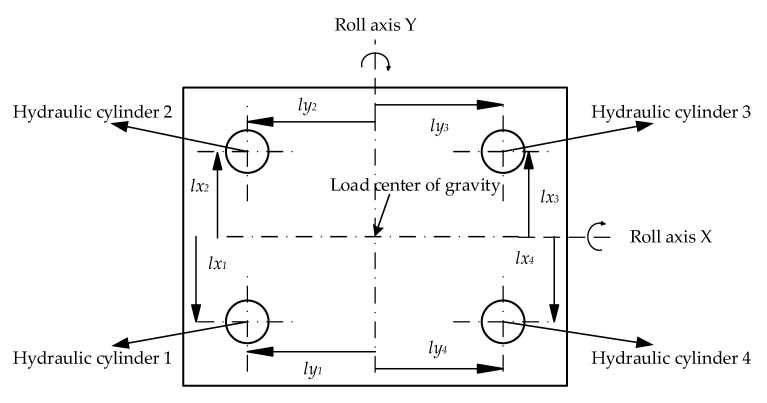
Distribution of contact points between four hydraulic cylinders and load.

**Figure 3 sensors-23-04092-f003:**
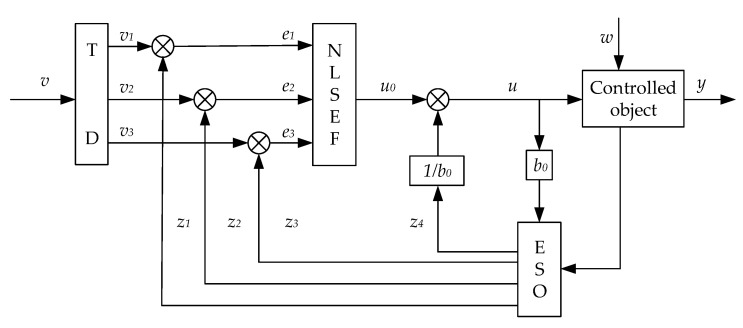
Structure of active disturbance rejection controller.

**Figure 4 sensors-23-04092-f004:**
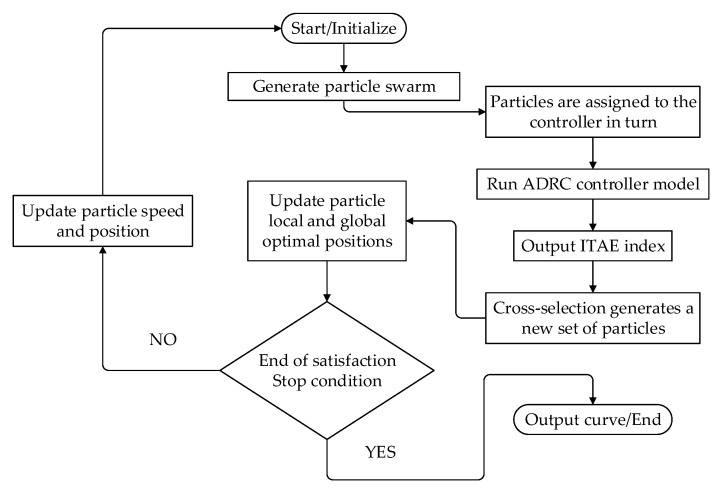
Improved particle swarm algorithm flow chart.

**Figure 5 sensors-23-04092-f005:**
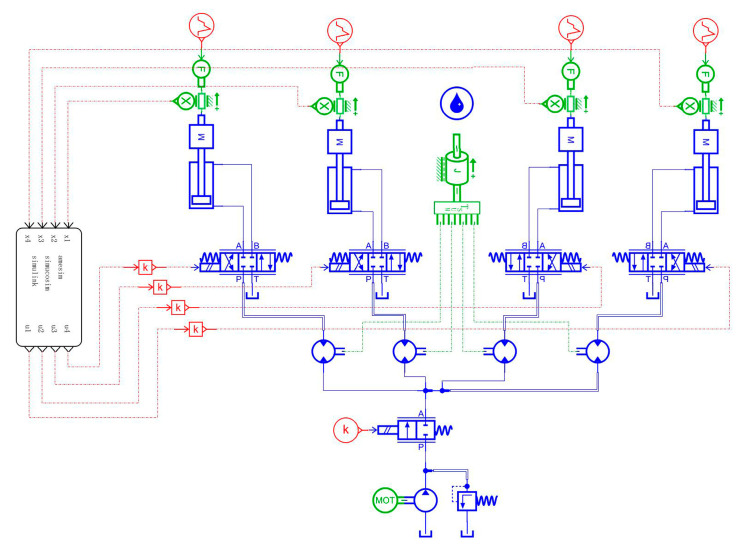
AMESim Physical Simulation Model for Multi Hydraulic Cylinder Group System.

**Figure 6 sensors-23-04092-f006:**
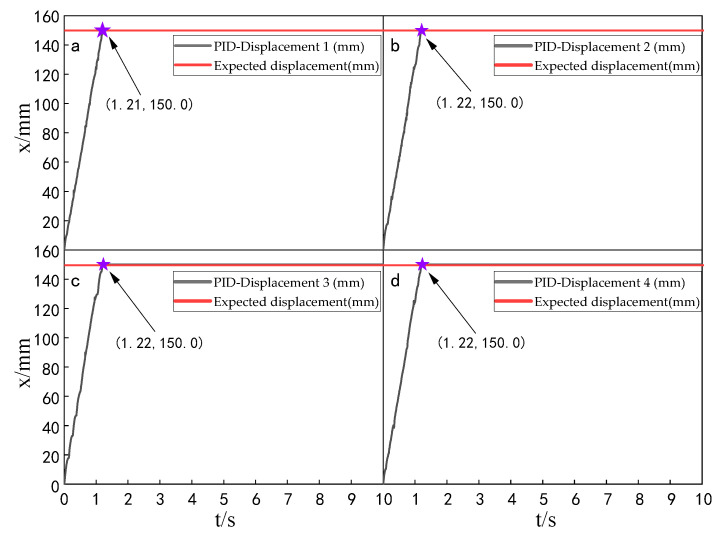
Displacement tracking curve of each cylinder under PID control algorithm when step signal is given.

**Figure 7 sensors-23-04092-f007:**
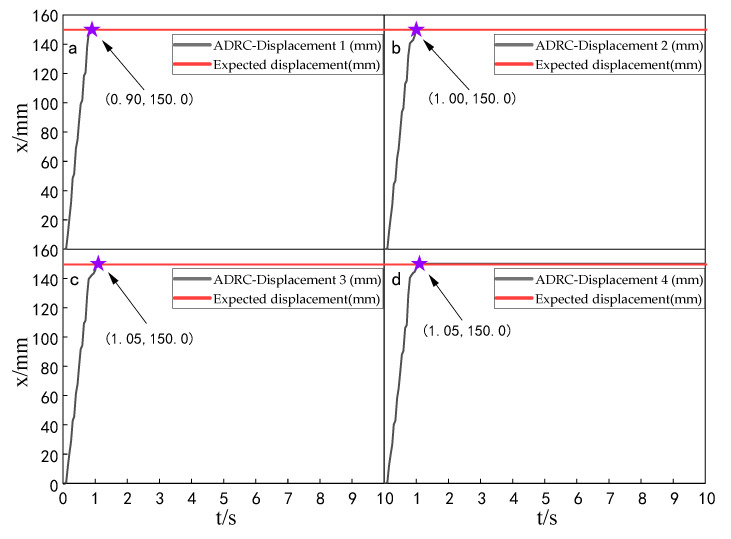
Displacement tracking curve of each cylinder under ADRC control algorithm when step signal is given.

**Figure 8 sensors-23-04092-f008:**
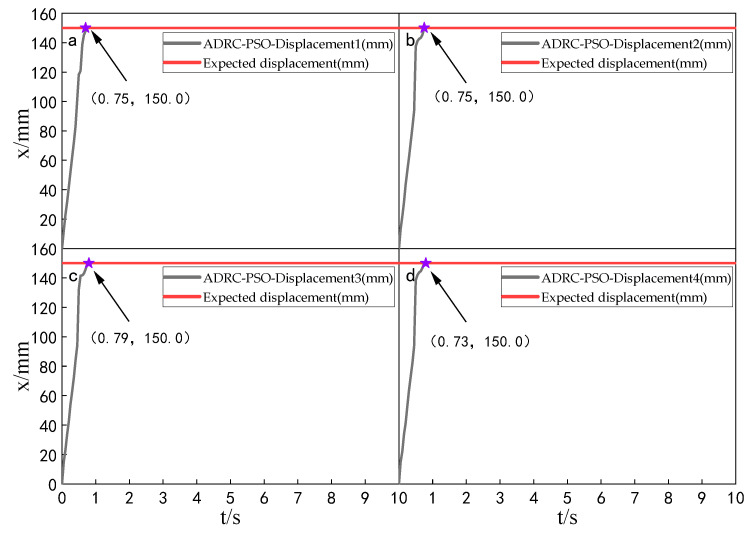
Displacement tracking curve of each cylinder under ADRC-PSO control algorithm when step signal is given.

**Figure 9 sensors-23-04092-f009:**
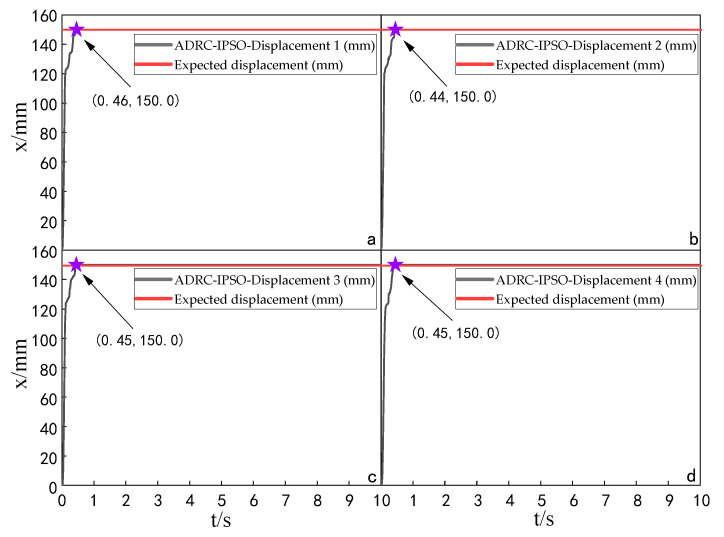
Displacement tracking curve of each cylinder under ADRC-IPSO control algorithm when step signal is given.

**Figure 10 sensors-23-04092-f010:**
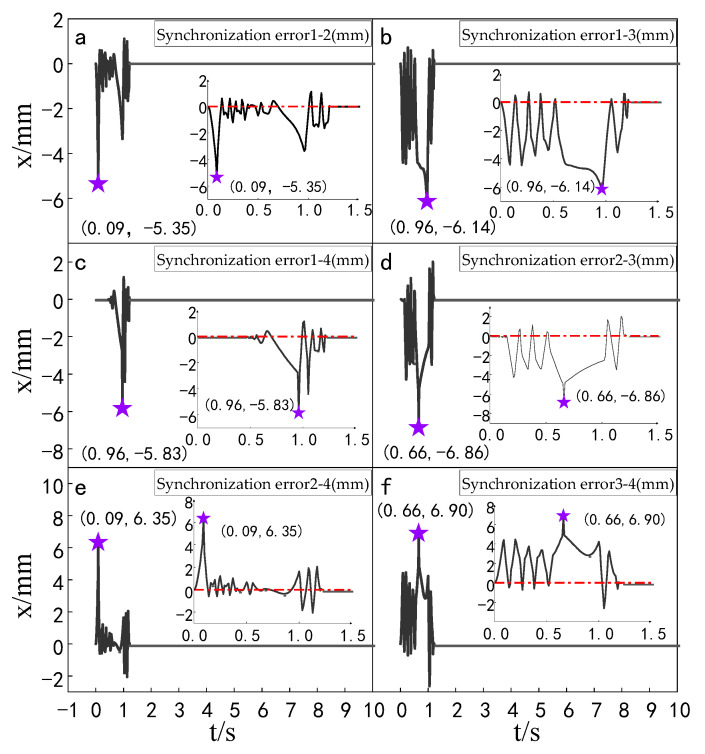
Synchronization error of each cylinder under PID control algorithm when step signal is given.

**Figure 11 sensors-23-04092-f011:**
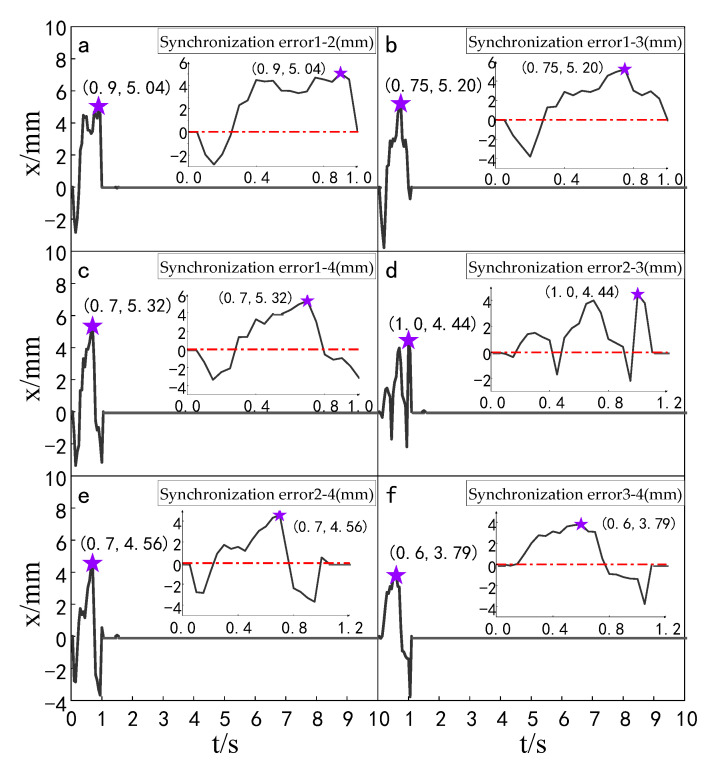
Synchronization error of each cylinder under ADRC control algorithm when step signal is given.

**Figure 12 sensors-23-04092-f012:**
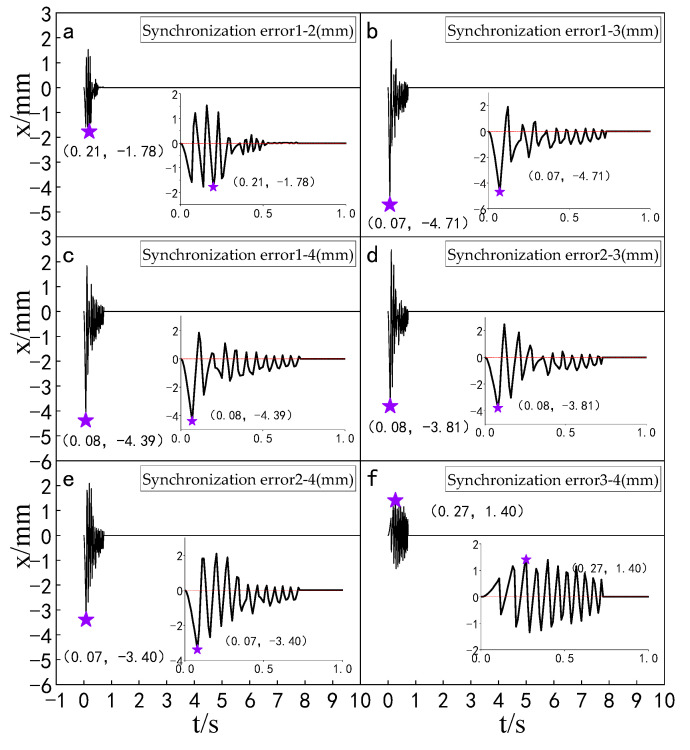
Synchronization error of each cylinder under ADRC-PSO control algorithm when step signal is given.

**Figure 13 sensors-23-04092-f013:**
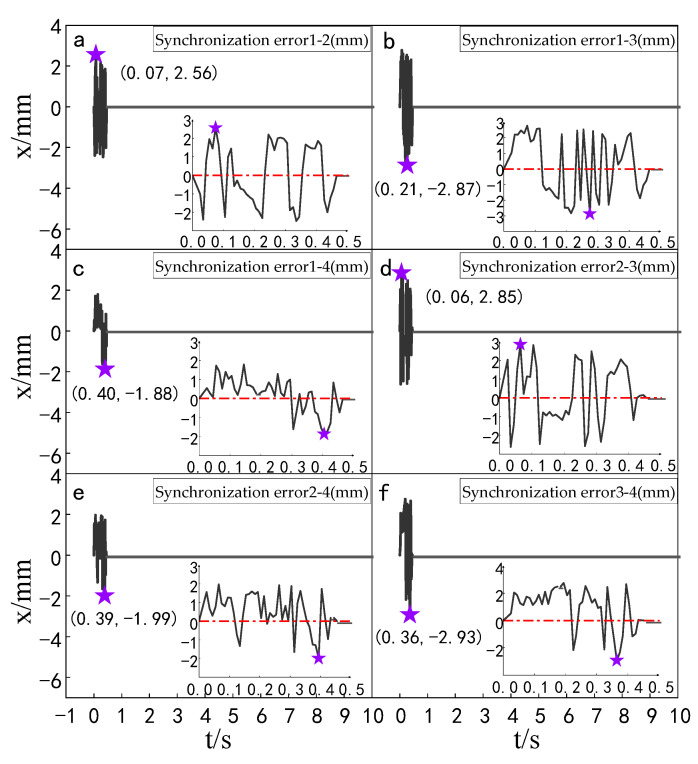
Synchronization error of each cylinder under ADRC-IPSO control algorithm when step signal is given.

**Figure 14 sensors-23-04092-f014:**
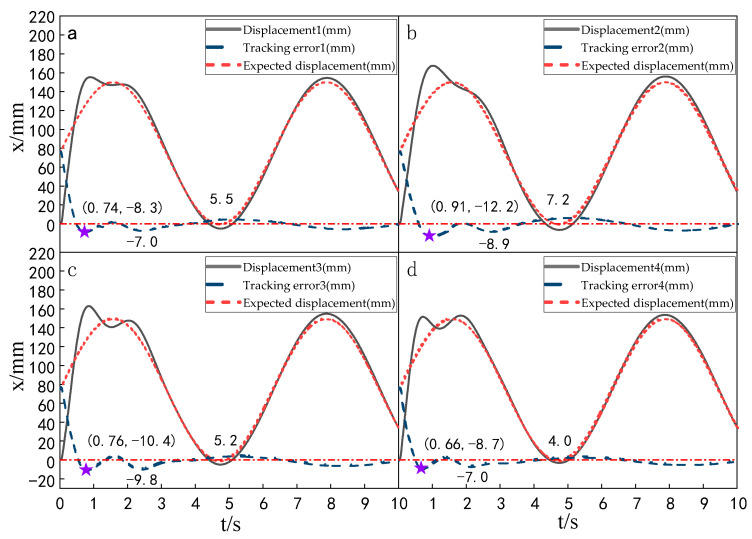
Displacement tracking curve of each cylinder calculated by the PID control when a sinusoidal signal is given.

**Figure 15 sensors-23-04092-f015:**
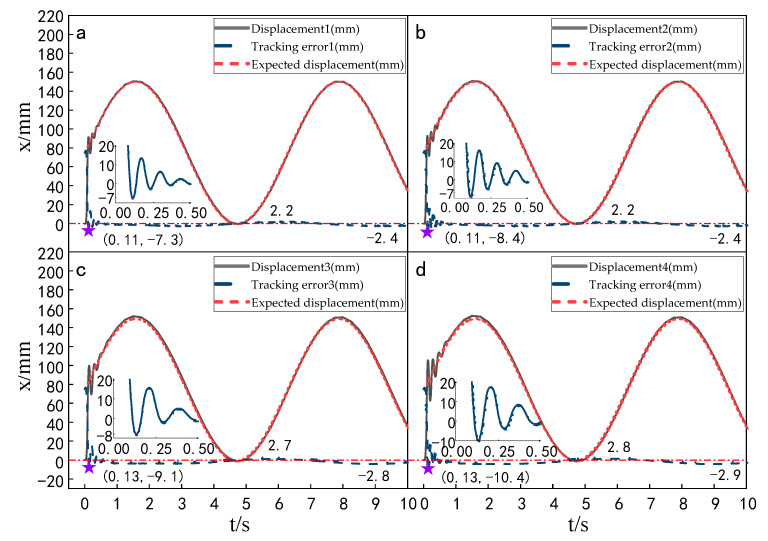
Displacement tracking curve of each cylinder calculated by the ADRC control when a sinusoidal signal is given.

**Figure 16 sensors-23-04092-f016:**
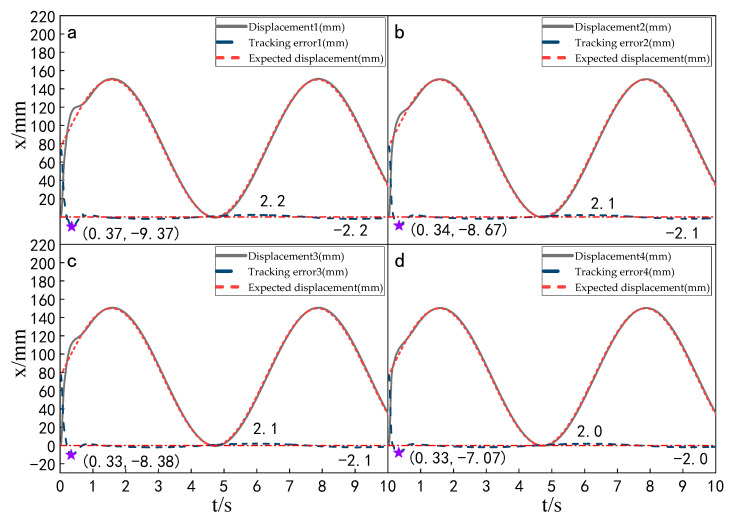
Displacement tracking curve of each cylinder calculated by the ADRC-PSO control when a sinusoidal signal is given.

**Figure 17 sensors-23-04092-f017:**
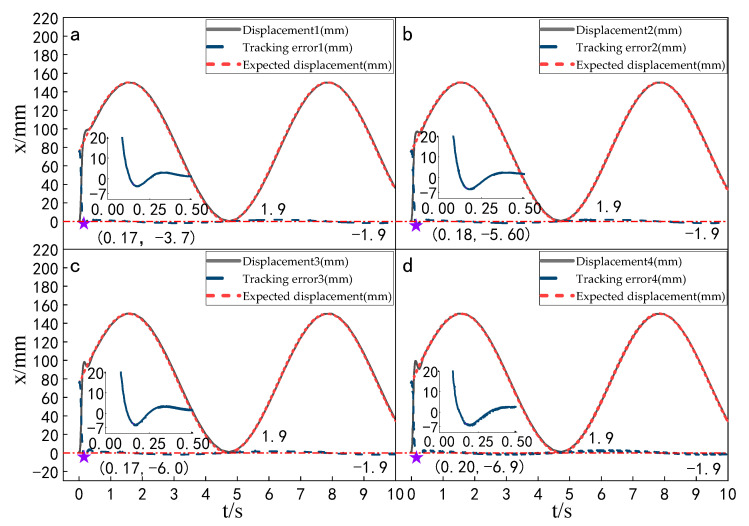
Displacement tracking curve of each cylinder calculated by ADRC-IPSO control when a sinusoidal signal is given.

**Figure 18 sensors-23-04092-f018:**
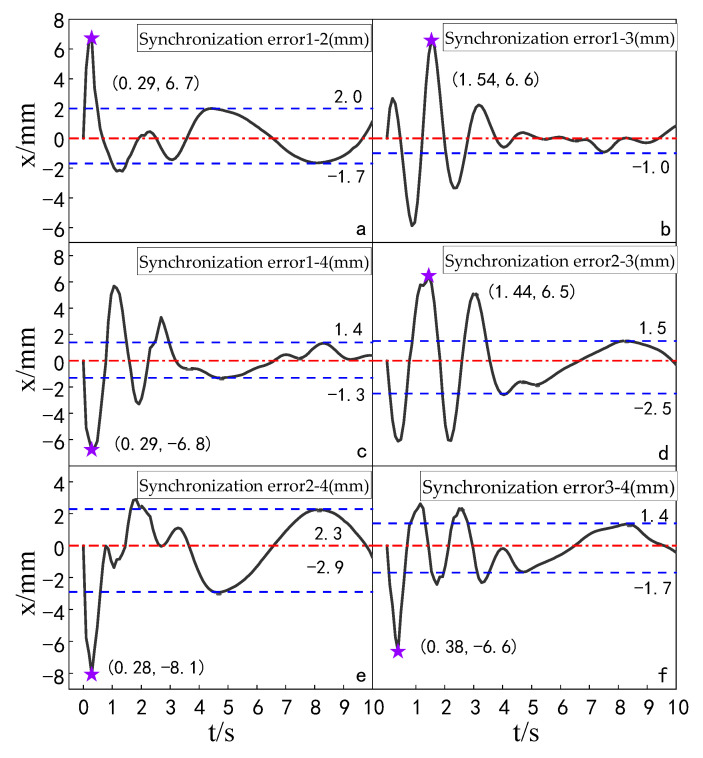
Synchronization error of each cylinder under the PID control algorithm when a sinusoidal signal is given.

**Figure 19 sensors-23-04092-f019:**
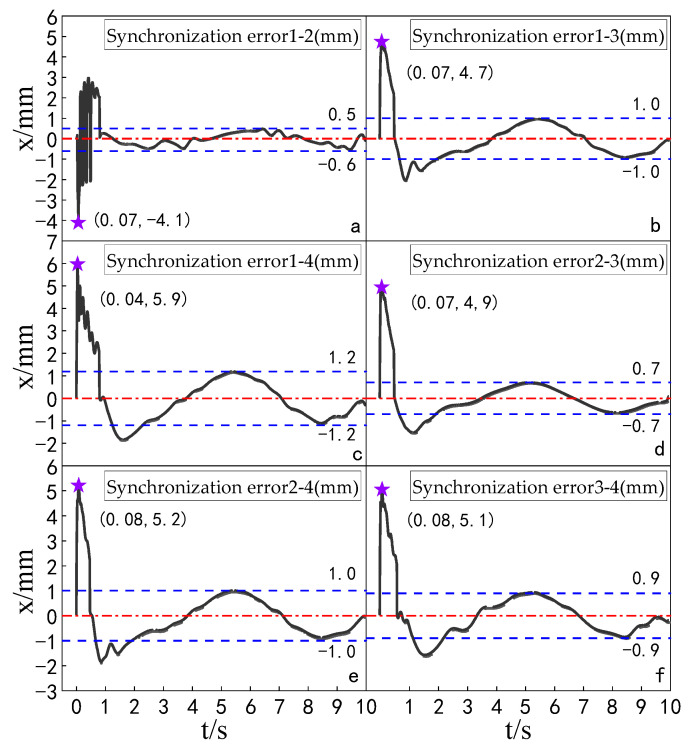
Synchronization error of each cylinder under the ADRC control algorithm when a sinusoidal signal is given.

**Figure 20 sensors-23-04092-f020:**
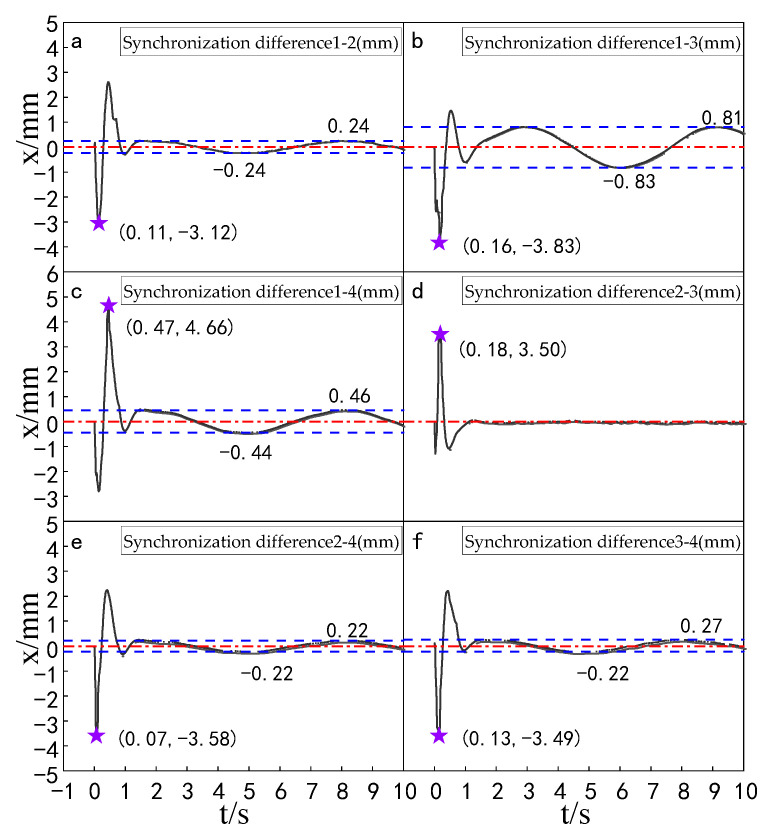
Synchronization error of each cylinder under the ADRC-PSO control algorithm when a sinusoidal signal is given.

**Figure 21 sensors-23-04092-f021:**
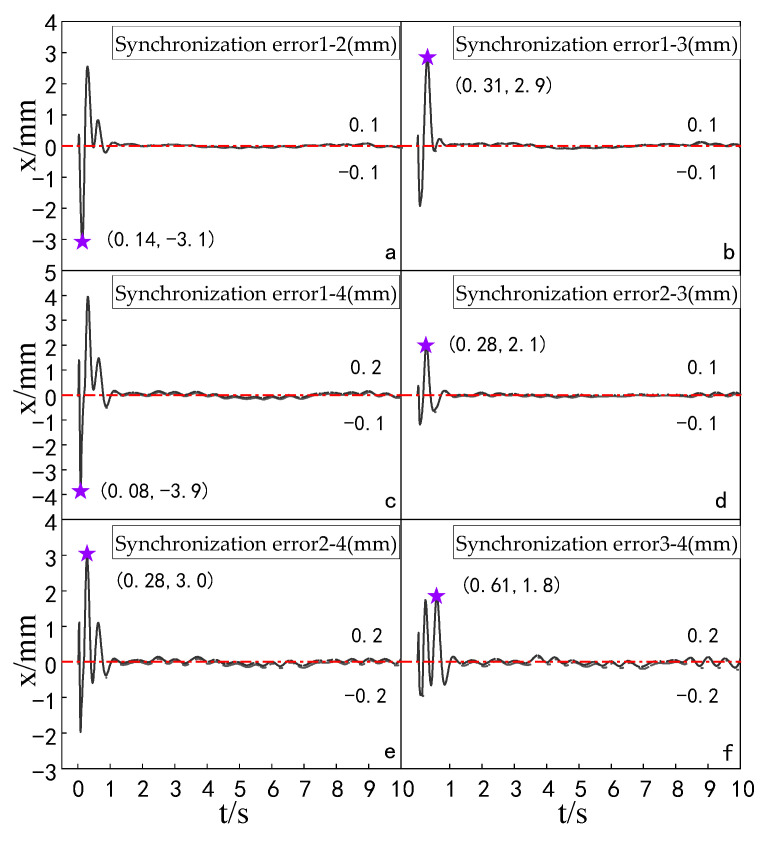
Synchronization error of each cylinder under the ADRC-IPSO control algorithm when a sinusoidal signal is given.

**Figure 22 sensors-23-04092-f022:**
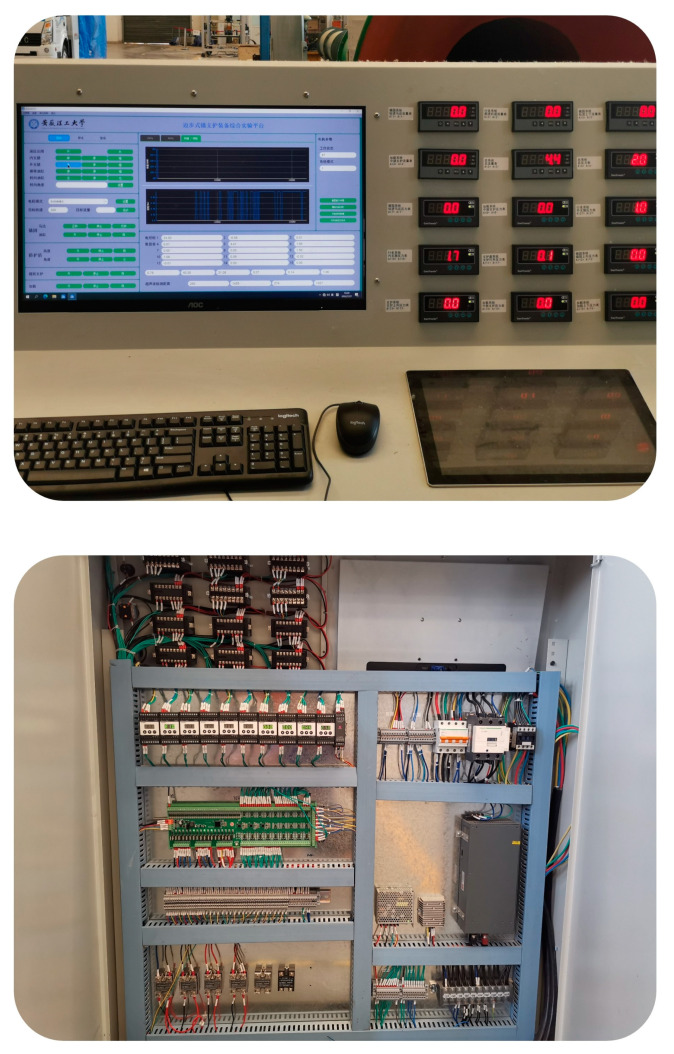
Electric control system and operation platform for multi-hydraulic cylinder group equipment of the digging-anchor-support robot.

**Figure 23 sensors-23-04092-f023:**
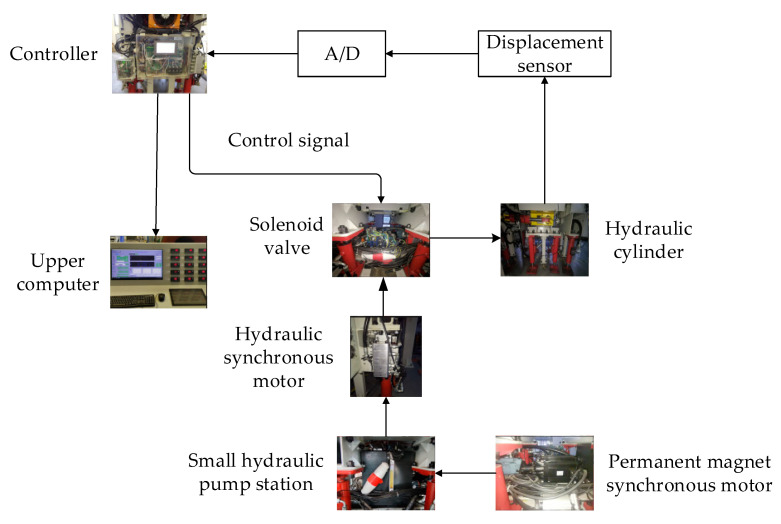
The experimental flow of active disturbance rejection control for multiple hydraulic cylinders of the digging-anchor-support robot.

**Figure 24 sensors-23-04092-f024:**
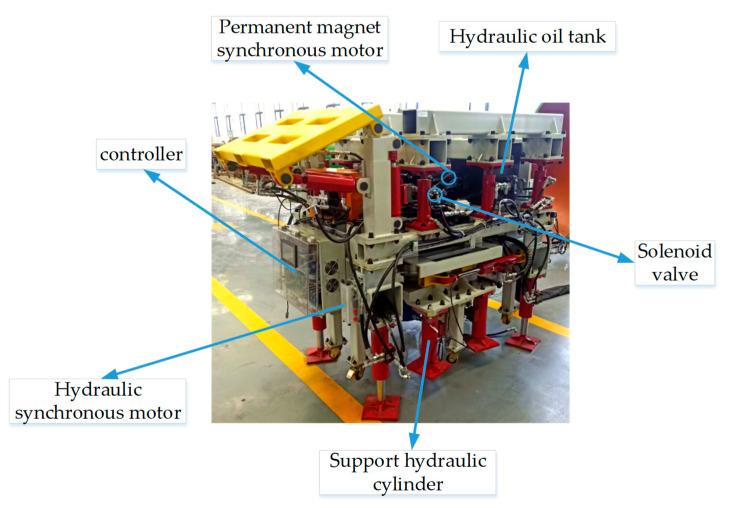
The physical platform of the multi-hydraulic cylinder group of the digging-anchor-support robot.

**Figure 25 sensors-23-04092-f025:**
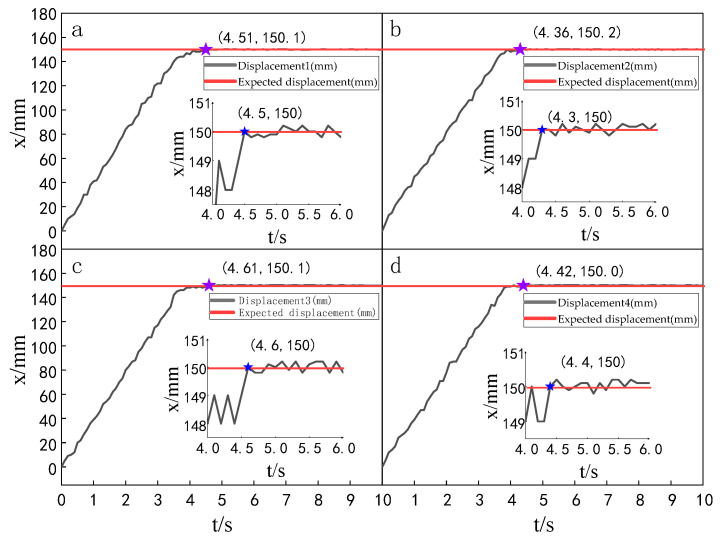
Displacement tracking curve of each cylinder under the PID control algorithm.

**Figure 26 sensors-23-04092-f026:**
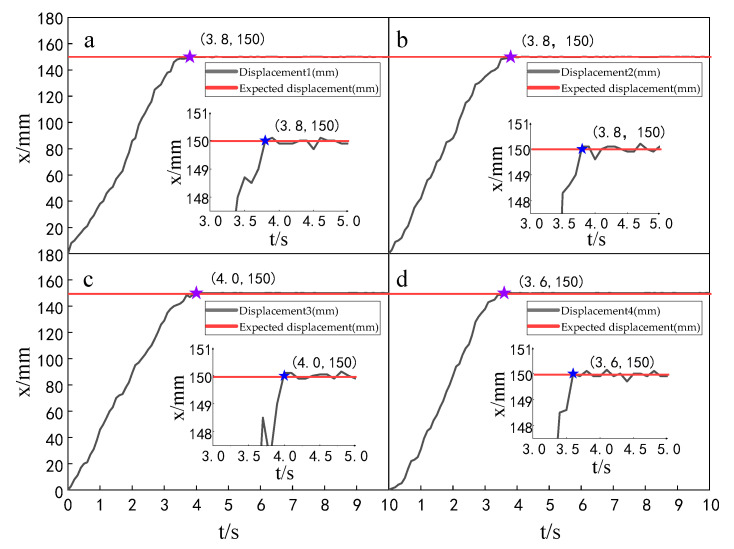
Displacement tracking curve of each cylinder under the ADRC control algorithm.

**Figure 27 sensors-23-04092-f027:**
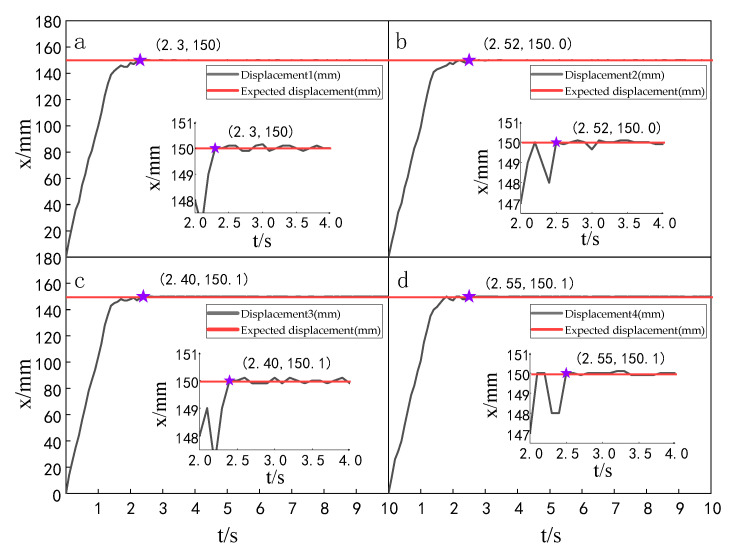
Displacement tracking curve of each cylinder under the ADRC-IPSO control algorithm.

**Figure 28 sensors-23-04092-f028:**
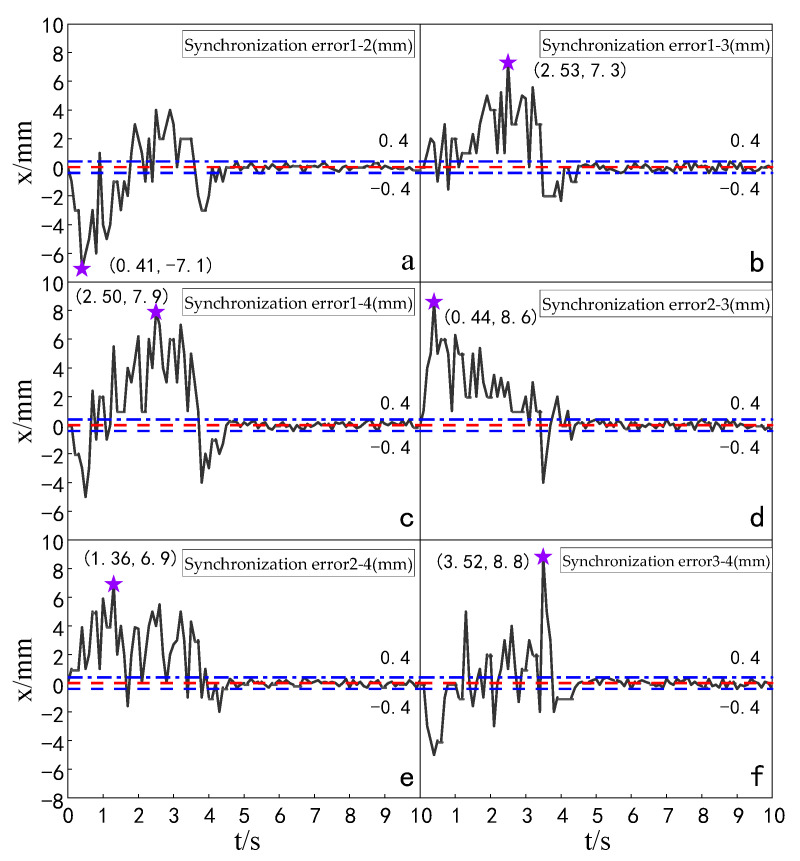
Synchronization error of each cylinder under the PID control algorithm.

**Figure 29 sensors-23-04092-f029:**
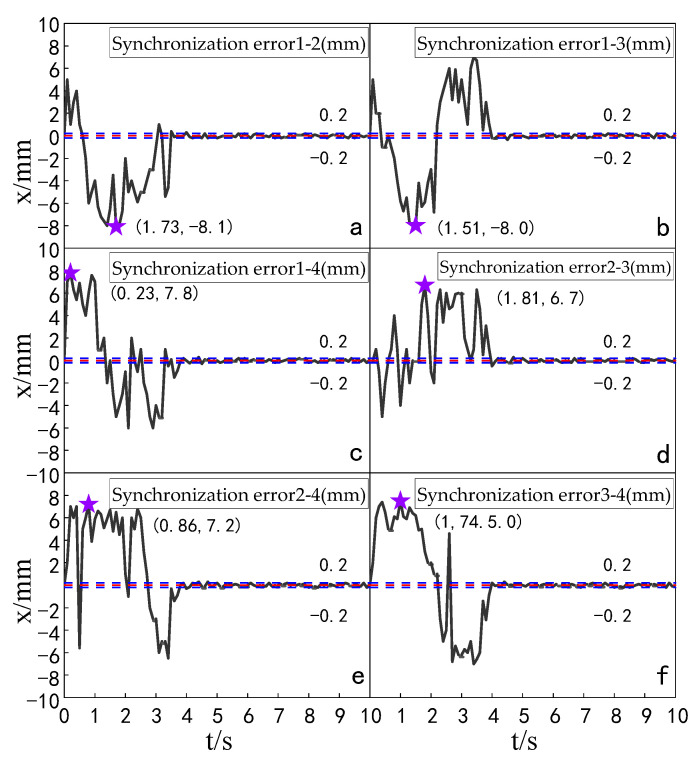
Synchronization error of each cylinder under the ADRC control algorithm.

**Figure 30 sensors-23-04092-f030:**
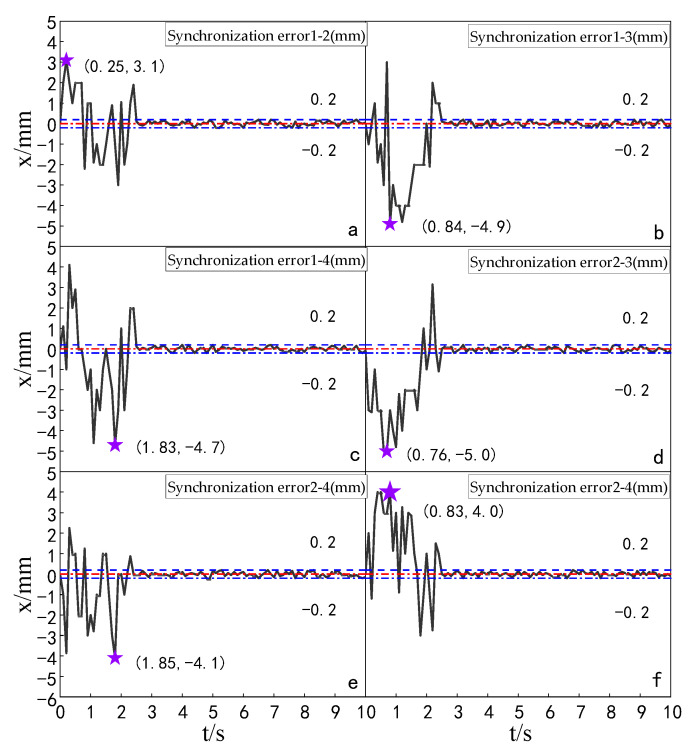
Synchronization error of each cylinder under the ADRC-IPSO control algorithm.

**Table 1 sensors-23-04092-t001:** Simulation expected signal parameters.

Name	Step Signal	Sinusoidal Signal	
Amplitude	150 mm	Frequency	1 rad/s
		Amplitude	150 mm

**Table 2 sensors-23-04092-t002:** Comparison of displacement tracking performance of each cylinder under different control algorithms when the step signal is given.

Controller	Maximum Adjustment Time (s)	Minimum Adjustment Time (s)	Steady State Value (mm)
PID	1.22	1.21	150.0
ADRC	1.05	0.90	150.0
ADRC-PSO	0.79	0.73	150.0
ADRC-IPSO	0.46	0.44	150.0

**Table 3 sensors-23-04092-t003:** Comparison of synchronization errors of each cylinder under different control algorithms when the step signal is given.

Controller	Maximum Synchronization Error (mm)	Minimum Synchronization Error (mm)	Steady State Error (mm)
PID	6.90	−5.35	0.00
ADRC	5.32	3.79	0.00
ADRC-PSO	−4.71	1.40	0.00
ADRC-IPSO	−2.93	−1.88	0.00

**Table 4 sensors-23-04092-t004:** Comparison of displacement tracking performance of each cylinder under different control algorithms when a sinusoidal signal is given.

Controller	Maximum Fluctuation Error (mm)	Maximum Amplitude Error (mm)	Maximum Adjustment Time (s)	Minimum Adjustment Time (s)
PID	−12.2	−9.8	1.88	1.33
ADRC	−10.4	−2.9	0.75	0.37
ADRC-PSO	−9.37	−2.2	0.68	0.49
ADRC-IPSO	−6.9	−1.9	0.37	0.29

**Table 5 sensors-23-04092-t005:** Comparison of synchronization errors of each cylinder under different control algorithms when a sinusoidal signal is given.

Controller	Maximum Synchronization Error (mm)	Minimum Synchronization Error (mm)	Stability Error (mm)
PID	−8.1	6.5	−2.9
ADRC	5.9	4.7	1.2
ADRC-PSO	4.66	−3.12	0.81
ADRC-IPSO	−3.9	2.1	0.2

**Table 6 sensors-23-04092-t006:** Experimental platform parameters.

Parameter Name	Value
Damping coefficient of hydraulic cylinder (N·S/m)	500
Leakage coefficient of hydraulic cylinder m5/(N·S)	1000
Natural frequency of electro-hydraulic proportional valve (rad/s)	80
Natural frequency of hydraulic cylinder (rad/s)	951.5
Damping ratio of electro-hydraulic proportional valve	0.7
Electro-hydraulic proportional valve current input (mA)	4~20
Equivalent mass of platform (kg)	900
Fluid density kg/m^3^	850
Bulk elastic modulus of oil (MPa)	1700
discharge coefficient (m^2^/s)	0.61
Supply pressure (MPa)	15
Feedback sensor gain	1
Plunger area (m^2^)	0.005
Rodless cavity area (mm^2^)	5024
Rod cavity area (mm^2^)	3061.5
Hydraulic cylinder stroke (mm)	150
Inner diameter of hydraulic cylinder (mm)	80
Piston rod diameter of hydraulic cylinder (mm)	50
Motor shaft speed (rev/min)	1500

**Table 7 sensors-23-04092-t007:** ADRC-IPSO controller parameters.

Parameter	Value	Parameter	Value
r	30	$a_{03}$	0.125
h	0.01	$\beta_{11}$	24,440
$h_{0}$	0.1	$\beta_{12}$	2.81
$\beta_{01}$	151.04	$\beta_{13}$	0.075
$\beta_{02}$	8554.91	$a_{11}$	−0.6
$\beta_{03}$	215,355.49	$a_{12}$	0.6
$\beta_{04}$	2,032,955.83	$a_{13}$	1.2
$a_{01}$	0.5	$b_{0}$	1.16
$a_{02}$	0.25	σ	0.01

**Table 8 sensors-23-04092-t008:** Comparison of displacement tracking performance of each cylinder under different control algorithms.

Controller	Maximum Adjustment Time (s)	Minimum Adjustment Time (s)	Steady State Value (mm)
PID	4.61	4.36	150.2
ADRC	4.06	3.63	150.1
ADRC-IPSO	2.55	2.31	150.0

**Table 9 sensors-23-04092-t009:** Comparison of cylinder synchronization errors under different control algorithms.

Controller	Maximum Synchronization Error (mm)	Minimum Synchronization Error (mm)	Steady State Error (mm)
PID	8.8	6.9	0.4
ADRC	−8.1	5.0	0.2
ADRC-IPSO	−5.0	3.1	0.2

## Data Availability

The data that support the findings of this study are available from the corresponding author upon reasonable request.

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
