# Peer review of "Research on Synchronous Control of Active Disturbance Rejection Position of Multiple Hydraulic Cylinders of Digging-Anchor-Support Robot"

_sensors, 2023, doi:10.3390/s23084092_

Round 1

Reviewer 1 Report

I have gone through your manuscript, and I do not have any critical comments to make. I believe it is well-written and well-organized as it is.

Author Response

It's a pleasure to hear from you. On behalf of all the authors of this article, I would like to thank you for your approval and wish you every success.

Reviewer 2 Report

The modelling section is not complete. There is no plant model. I.e. how do the valve commands, u, relate the to cylinder forces or velocities? The results seem good, but without these equations and parameters it is impossible to evaluate the controller or whether the results are plausible.

Reviewer 3 Report

This paper proposed an improved automatic disturbance rejection controller- improved particle swarm optimization position synchronization control method.

To verify the reliability of the ADRC-IPSO control algorithm, authors compared with the traditional auto-disturbance rejection control algorithm and the classical PID control algorithm, in order to reflect the degree of improvement, the performance comparison of traditional particle swarm optimization algorithms should also be added into 4. Simulation analysis part.

The joint simulation is carried out by using MATLAB and AMESim software, is it a hydraulic cylinder or an integral mechanism? Joint parameters need to be given, specific parameters such as the size of the large and small cavities, the length of the hydraulic cylinder, and the oil supply pressure. Hydraulic AMESim simulation models and parameters also need to be given.

Steady state value is 150.0, why is there no overshoot under PID control and what are the parameters of PID controller?

In the experimental part, in addition to considering the controllers differences from the perspective of control accuracy, control quantities is also an indicator to visually express the reliability.

Experimental platform need to be described in more detail, such as sensor parameters and types.

Round 2

Reviewer 3 Report

The author has made corresponding changes, this work can meet requirements.